# Uncovering the Structural Fairness in Graph Contrastive Learning

**Ruijia Wang[1], Xiao Wang[1]***, **Chuan Shi[1]***, **Le Song[2]**
[1]Beijing University of Posts and Telecommunications
[2] BioMap and MBZUAI
{wangruijia, xiaowang, shichuan}@bupt.edu.cn, songle@biomap.com

## Abstract

Recent studies show that graph convolutional network (GCN) often performs worse for low-degree nodes, exhibiting the so-called structural unfairness for graphs with long-tailed degree distributions prevalent in the real world. Graph contrastive learning (GCL), which marries the power of GCN and contrastive learning, has emerged as a promising self-supervised approach for learning node representations. How does GCL behave in terms of structural fairness? Surprisingly, we find that representations obtained by GCL methods are already fairer to degree bias than those learned by GCN. We theoretically show that this fairness stems from intra-community concentration and inter-community scatter properties of GCL, resulting in a much clear community structure to drive low-degree nodes away from the community boundary. Based on our theoretical analysis, we further devise a novel graph augmentation method, called GRAph contrastive learning for DEgree bias (GRADE), which applies different strategies to low- and high-degree nodes. Extensive experiments on various benchmarks and evaluation protocols validate the effectiveness of the proposed method.

## 1 Introduction

Despite their strong expressive power in graph representation learning, recent studies reveal that the performance of vanilla graph convolutional network (GCN) [17] exhibits a structural unfairness [25, 16], which is primarily beneficial to high-degree nodes (head nodes) but biased against low-degree nodes (tail nodes). Such a performance disparity is alarming and causes a performance bottleneck, given that node degrees of real-world graphs often follow a long-tailed power-law distribution [2].

Graph contrastive learning (GCL) [30, 24, 22, 38] has been a promising paradigm in graph domain, which integrates the power of GCN and contrastive leaning [13, 5, 10]. In a nutshell, these methods typically construct multiple views via stochastic augmentations of the input, then optimize the GCN encoder by contrasting positive samples against negative ones. Inheriting advantages of contrastive learning, GCL relieves graph representation learning from human annotations, and displays state-of-the-art performance in a variety of tasks [35, 12, 39, 36].

*Will GCL present the same structure unfairness as GCN*? For this purpose, we conduct experiments to surprisingly find out that GCL methods are better at maintaining structural fairness where a smaller performance gap exists between tail nodes and head nodes than that of GCN. This finding suggests that GCL has the potential to mitigate structural unfairness. Based on this finding, a natural and fundamental question arises: *why is graph contrastive learning fairer to degree bias?* A well-informed answer can yield profound insights into solutions to this important problem, and deepen our understanding of the mechanism of GCL.

---

*Corresponding authors.

36th Conference on Neural Information Processing Systems (NeurIPS 2022).

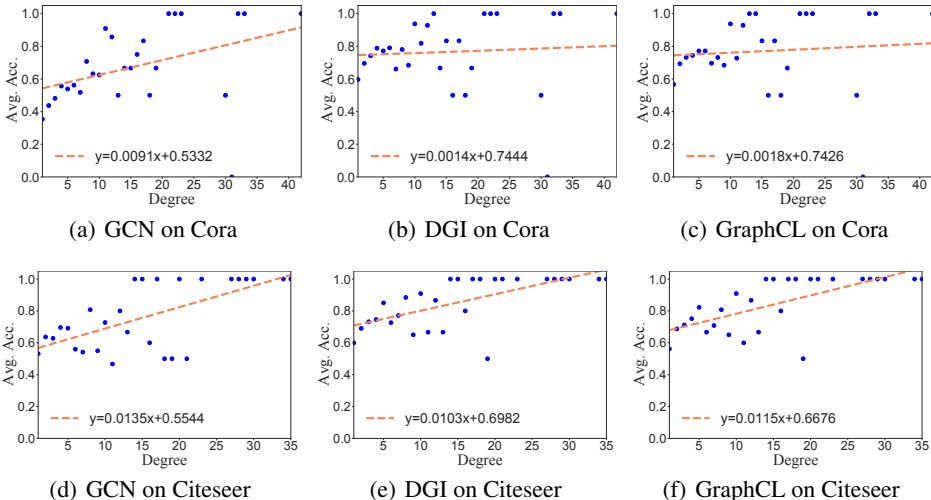

Figure 1: Visualization for the fairness of models to degree bias. Blue scatters refer to the average accuracy (Avg. Acc.) of a specific degree group. Orange dotted lines are regression lines of blue scatters in each figure. For clarity, we annotate analytical expressions of regression lines. The higher the average accuracy of tail nodes and the smaller the slope of the regression line indicate that the model is fairer to degree bias.

Intuitively, graph augmentation provides an opportunity for tail nodes to generate more within-community edges, making their representations closer to those with the same community via the contrastive framework. These refined representations drive tail nodes away from the community boundary. Theoretically, we prove that node representations learned by GCL conform to a clearer community structure by Intra-community Concentration Theorem 1 and Inter-community Scatter Theorem 2. These theorems are relevant to two crucial components in GCL. One is the alignment of positive pairs, which is exactly the optimization objective. The other is the pre-defined graph augmentation, determining the concentration of augmented representations. Based on the analysis, we establish the relation between graph augmentation and representation concentration, implying that a well-designed graph augmentation can promote structural fairness by concentrating augmented representations.

To take a step further, we propose a GRAph contrastive learning for DEgree bias (GRADE) based on a novel graph augmentation. To make augmented representations more concentrated within communities, GRADE enlarges limited neighbors of tail nodes to contain more nodes within the same community, where the ego network of the tail node interpolates with that of the sampled similar node. As for head nodes, GRADE purifies their ego networks by removing neighbors from different communities. Extensive experiments on various benchmark datasets and several evaluation protocols validate the effectiveness of GRADE.

In summary, our contributions are four-fold:

- We are the first to discover that GCL methods exhibit more structural fairness than GCN, which has a smaller performance disparity between tail nodes and head nodes. This discovery inspires a new path for alleviating structural unfairness based on contrastive learning.

- We theoretically validate the reason for structural fairness in GCL is that GCL stimulates intra-community concentration and inter-community scatter. Therefore, tail nodes are farther away from the community boundary for better classification.

- Based on theoretical insights, we propose a method GRADE to further improve structural fairness by enriching the neighborhood of tail nodes while purifying neighbors of head nodes.

- Comprehensive experiments demonstrate that our GRADE outperforms baselines on multiple benchmark datasets and enhances the fairness to degree bias.

## 2 Exploring the Behavior of Graph Contrastive Learning on Degree Bias

Real-world graphs in many domains follow a long-tailed distribution in node degrees, i.e., a significant fraction of nodes are tail nodes with small degrees. It is well known that GCN often performs worse accuracy for tail nodes. Here we aim to study how GCL behaves under degree bias.

**Experimental Setup**  We take two representative GCL algorithms DGI [30] and GraphCL [35] as examples to analyze the performance disparity under degree bias. Specifically, we train DGI and GraphCL on four benchmarks Cora [17], Citeseer [17], Photo [23] and Computer [23], and leverage early stopping based on the training loss. To compare with GCN, our linear evaluation protocol deploys the semi-supervised split [17], where 20 labeled nodes per class form training set and test set composes of randomly sampled 1000 nodes with degrees less than 50. GCN follows the standard training paradigm [17] with the above train-test split. Further implementation details and chosen hyper-parameters are deferred to Appendix A.

**Results**  To illustrate the fairness to degree bias, we group nodes of the same degree and calculate the average accuracy of these degree groups separately shown in Figure 1. To further visualize how the model balances the performance between tail nodes and head nodes, we fit these scatters with linear regression. If the slope of the regression line is flat, the model is fair to degree bias. More results on Photo and Computer datasets can be seen in Appendix A. From the figure, we can find that the average accuracy for tail nodes of GCL methods DGI and GraphCL is higher than that of GCN, and the slope of the regression line is also smaller. These interesting observations suggest that label-independent GCL methods are actually fairer than GCN under degree bias.

## 3 Analysis on the Structural Fairness of Graph Contrastive Learning

Based on the above observations, a natural and fundamental question arises: where does this structural fairness stem from? We first define some preliminary notations, then provide a theoretical analysis to explain this question.

### 3.1 Preliminary Notations

Let $G = (\mathcal{V}, \mathcal{E}, X)$ be a graph, where $\mathcal{V}$ is the set of $N$ nodes $\{v_1, v_2, \cdots, v_N\}$, $\mathcal{E} \subseteq \mathcal{V} \times \mathcal{V}$ is the set of edges, $X = [\boldsymbol{x}_1, \boldsymbol{x}_2, \cdots, \boldsymbol{x}_N] \in \mathbb{R}^{N \times B}$ represents the node feature matrix and $\boldsymbol{x}_i$ is the feature vector of node $v_i$. The edges can be represented by an adjacency matrix $A \in \{0, 1\}^{N \times N}$, where $A_{ij} = 1$ iff $(v_i, v_j) \in \mathcal{E}$. Given unlabeled training nodes, each node belongs to one of $K$ latent communities $C_1, C_2, \cdots, C_K$. Assuming the augmentation set $\mathcal{T}$ consisting of all transformations on topology, the set of potential positive samples generated from ego network $\mathcal{G}_i$ of node $v_i$ is denoted as $\mathcal{T}(\mathcal{G}_i)$. The goal of GCL is to learn a GCN encoder $f$ such that positive pairs are closely aligned while negative pairs are far apart. Here we focus on topological augmentation and single-layer GCN,

$$f(\mathcal{G}_i) = ReLU(\tilde{L}_i X W), \tag{1}$$

where $\tilde{L}_i$ is the $i$-row of transition matrix $\tilde{L} = \tilde{D}^{-1}\tilde{A}$, $\tilde{A} = A + I$ is self-looped adjacency matrix and $\tilde{D}_{ii} = \sum_j \tilde{A}_{ij}$ is degree matrix. We consider a community indicator $F_f$

$$F_f(\mathcal{G}_i) = \underset{k \in [K]}{\arg\min} \| f(\mathcal{G}_i) - \mu_k \|, \tag{2}$$

where $\mu_k = \mathbb{E}_{v_i \in C_k} \mathbb{E}_{\hat{\mathcal{G}}_i \in \mathcal{T}(\mathcal{G}_i)}[f(\hat{\mathcal{G}}_i)]$ is the community center, and $\| \cdot \|$ stands for $l_2$-norm. To quantify the performance of $F_f$, the error can be formulated as

$$\text{Err}(F_f) = \sum_{k=1}^{K} \mathbb{P}[F_f(\mathcal{G}_i) \neq k, \forall v_i \in C_k]. \tag{3}$$

With the above definitions, we denote $S_\varepsilon = \{v_i \in \cup_{k=1}^{K} C_k : \forall \hat{\mathcal{G}}_i^1, \hat{\mathcal{G}}_i^2 \in \mathcal{T}(\mathcal{G}_i), \|f(\hat{\mathcal{G}}_i^1) - f(\hat{\mathcal{G}}_i^2)\| \leq \varepsilon\}$ as the set of nodes with $\varepsilon$-close representations among graph augmentations.

### 3.2 Theoretical Analysis

We assume the nonlinear transformation has $M$-Lipschitz continuity, i.e., $\|f(\mathcal{G}_i) - f(\mathcal{G}_j)\| = \|ReLU(\tilde{L}_i X W) - ReLU(\tilde{L}_j X W)\| \leq M \|\tilde{L}_i X - \tilde{L}_j X\|$, and graph augmentations are uniformly

sampled with $m$ augmented edges $\mathbb{P}[\hat{\mathcal{G}}_i = \mathcal{T}(\mathcal{G}_i)] = 1/C(N-1, m)$. Let there be a ball of radius $\beta m$ such that for any augmentation $\|\tilde{L}_i X - \hat{L}_i X\|^2 \leq \beta m$, where $\hat{L}$ is the augmented transition.

**Theorem 1** *Intra-community Concentration. Let pre-transformation representations $\tilde{L}X$ be sub-Gaussian random variable with variance $\sigma^2$. For all nodes $v_i \in S_\varepsilon$, if $\varepsilon^2 \leq \frac{\beta m}{6M^2\kappa}$, their representations $f(\mathcal{G}_i)$ fit sub-Gaussian distribution with variance $\sigma^2_{f,\varepsilon} \leq \frac{1}{\kappa}\sigma^2$ with $\kappa \geq 1$ where $\kappa$ is a coefficient that reflects the degree of concentration.*

This theorem builds a relation between the intra-community concentration of final representations and the alignment of positive pairs in $S_\varepsilon$. Specifically, intra-commmunity concentration requires smaller $\varepsilon$. By decreasing the distance between positive pairs, GCL fits the requirement.

Next, we demonstrate that GCL also maintains the property of inter-community scatter for community assignment. For a given augmentation set $\mathcal{T}$, we first define the augmentation distance between two nodes as the minimum distance between their pre-transformation representations,

$$d_\mathcal{T}(v_i, v_j) = \min_{\hat{\mathcal{G}}_i \in \mathcal{T}(\mathcal{G}_i), \hat{\mathcal{G}}_j \in \mathcal{T}(\mathcal{G}_j)} \|\hat{L}_i X - \hat{L}_j X\| = \min_{\hat{\mathcal{G}}_i \in \mathcal{T}(\mathcal{G}_i), \hat{\mathcal{G}}_j \in \mathcal{T}(\mathcal{G}_j)} \|(\frac{\hat{A}_i}{\hat{d}_i} - \frac{\hat{A}_j}{\hat{d}_j})X\|, \quad (4)$$

where $\hat{A}_i$ is the $i$-row of augmented adjacency matrix $\hat{A}$, and $\hat{d}_i$ is the augmented node degree. Based on the augmentation distance, we further introduce the definition of $(\alpha, \gamma, \hat{d})$-augmentation to measure the concentration of pre-transformation representations.

**Definition 1** *($\alpha, \gamma, \hat{d}$)-Augmentation. The augmentation set $\mathcal{T}$ is a $(\alpha, \gamma, \hat{d})$-augmentation, if for each community $C_k$, there exists a subset $C_k^0 \subset C_k$ such that the following two conditions hold*

    *1. $\mathbb{P}[v_i \in C_k^0] \geq \alpha\mathbb{P}[v_i \in C_k]$ where $\alpha \in (0,1]$,*

    *2. $\sup_{v_i, v_j \in C_k^0} d_\mathcal{T}(v_i, v_j) \leq \gamma(\frac{B}{\hat{d}_{\min}^k})^{\frac{1}{2}}$ where $\gamma \in (0,1]$,*

*where $\hat{d}_{\min}^k = \min_{v_i \in C_k^0, \hat{\mathcal{G}}_i \in \mathcal{T}(\mathcal{G}_i)} \hat{d}_i$, and $B$ is the feature dimension.*

Larger $\alpha$ and smaller $\gamma(B/\hat{d}_{\min}^k)^{\frac{1}{2}}$ indicate pre-transformation representations are more concentrated. We assume that the representation is normalized by $\|f(\mathcal{G}_i)\| = r$ and let $p_k = \mathbb{P}[v_i \in C_k]$. Then we simultaneously bound the inter-community distance and the error of the community indicator.

**Lemma 1** *For a $(\alpha, \gamma, \hat{d})$-augmentation with subset $C_k^0$ of each community $C_k$, if nodes belonging to $(C_1^0 \cup \cdots \cup C_K^0) \cap S_\varepsilon$ can be correctly assigned by the community indicator $F_f$, then the error of all nodes can be bounded by $(1 - \alpha) + R_\varepsilon$, where $R_\varepsilon = \mathbb{P}[\overline{S_\varepsilon}]$ is the proportion of complement.*

The above lemma presents a sufficient condition to guarantee the performance of the community indicator. Then we need to explore when nodes in $(C_1^0 \cup \cdots \cup C_K^0) \cap S_\varepsilon$ can be correctly assigned by $F_f$.

**Lemma 2** *For a $(\alpha, \gamma, \hat{d})$-augmentation and each $\ell \in [K]$, if*

$$\mu_\ell^\top \mu_k < r^2(1 - \rho_\ell(\alpha, \gamma, \hat{d}, \varepsilon) - \sqrt{2\rho_\ell(\alpha, \gamma, \hat{d}, \varepsilon)} - \frac{\Delta_\mu}{2})$$

*holds for all $k \neq \ell$, then every node $v_i \in C_\ell^0 \cap S_\varepsilon$ can be correctly assigned by the community indicator $F_f$, where $\rho_\ell(\alpha, \delta, \varepsilon) = 2(1 - \alpha) + \frac{2R_\varepsilon}{p_\ell} + \alpha(\frac{M\gamma\sqrt{B}}{r\sqrt{\hat{d}_{\min}^\ell}} + \frac{2\varepsilon}{r})$ and $\Delta_\mu = 1 - \min_{k \in [K]} \|\mu_k\|^2/r^2$.*

Combining Lemma 1 and 2, we can obtain the Inter-community Scatter Theorem as follows.

**Theorem 2** *Inter-community Scatter. For a $(\alpha, \gamma, \hat{d})$-augmentation, if*

$$\mu_\ell^\top \mu_k < r^2(1 - \rho_{\max}(\alpha, \gamma, \hat{d}, \varepsilon) - \sqrt{2\rho_{\max}(\alpha, \gamma, \hat{d}, \varepsilon)} - \frac{\Delta_\mu}{2}) \quad (5)$$

*holds for any pair of $(\ell, k)$ with $\ell \neq k$, then the error of the community indicator $F_f$ can be bounded by $(1 - \alpha) + R_\varepsilon$, where $\rho_{\max}(\alpha, \gamma, \hat{d}, \varepsilon) = 2(1 - \alpha) + \max_\ell \left( \frac{2R_\varepsilon}{p_\ell} + \frac{M\alpha\gamma\sqrt{B}}{r\sqrt{\hat{d}_{\min}^\ell}} \right) + \frac{2\alpha\varepsilon}{r})$ and $\Delta_\mu = 1 - \min_{k \in [K]} \|\mu_k\|^2 / r^2$.*

For the better assignment, the RHS of Equation 27 should approach $r^2$, implying that smaller $\rho_{\max}(\alpha, \gamma, \hat{d}, \varepsilon)$ is required. We further bound $R_\varepsilon$ by the alignment objective of contrastive loss.

**Theorem 3** *The term $R_\varepsilon$ is upper bounded by*

$$R_\varepsilon \leq \frac{[C(N-1, m)]^2}{\varepsilon} \mathbb{E}_{v_i} \mathbb{E}_{\hat{\mathcal{G}}_i^1, \hat{\mathcal{G}}_i^2 \in \mathcal{T}(\mathcal{G}_i)} \|f(\hat{\mathcal{G}}_i^1) - f(\hat{\mathcal{G}}_i^2)\|. \tag{6}$$

We direct the readers to Appendix B for proof of all the above lemmas and theorems. The combination of Theorem 2 and Theorem 3 indicates that the inter-community distance and the error of community indicator are controlled by two key factors. 1) The alignment of positive pairs. Good alignment enables small $\mathbb{E}_{\hat{\mathcal{G}}_i^1, \hat{\mathcal{G}}_i^2 \in \mathcal{T}(\mathcal{G}_i)} \|f(\hat{\mathcal{G}}_i^1) - f(\hat{\mathcal{G}}_i^2)\|$, resulting in small $R_\varepsilon$. 2) The concentration of augmented representations, where sharper concentration implies larger $\alpha$ in Definition 1. Small $R_\varepsilon$ and large $\alpha$ directly decrease the error bound of community indicator, and provide small $\rho_{\max}(\alpha, \gamma, \hat{d}, \varepsilon)$ for inter-community scatter. It is worth mentioning that the first factor is the contrastive objective in GCL, reflecting the reason for structural fairness in the GCL framework. While the second factor depends on the graph augmentation. Thus, we are motivated to propose a graph augmentation designed for further concentrating augmented representations.

## 4 GRADE Methodology

In this section, we present our novel GRADE framework dedicated to degree bias in detail, starting with the special graph augmentation, followed by the objective of GCL.

### 4.1 Graph Augmentation

We generate two augmentations $\hat{G}_1$ and $\hat{G}_2$ by simultaneously corrupting the original feature and topology to construct diverse contexts [36, 26]. We denote node representations in these two augmentations as $H = f(\hat{G}_1)$ and $O = f(\hat{G}_2)$.

**Topology Augmentation**   To obtain more concentrated augmented representations, we aim to increase intra-community edges while decreasing inter-community edges. Due to the different structural properties of tail nodes and head nodes, we design different topology augmentation strategies for them shown in Figure 2. In order to expand the neighborhood of tail nodes to include more same-community nodes, we interpolate the ego network of the anchor tail node $v_{tail}$ with that of a sampled similar node $v_{sample}$. To prevent injecting many different-community neighbors, we regulate the interpolation ratio depending on the similarity between $v_{tail}$ and $v_{sample}$. For head nodes, we purify their neighborhood by similarity-based sampling to remove inter-community edges.

Formally, we build the similarity matrix $S$ between node pairs based on cosine similarity of their representations, $S_{ij} = \text{sim}(\boldsymbol{h}_i, \boldsymbol{h}_j)$ for $i \neq j$ and $S_{ii} = 0$ otherwise. For any tail node $v_{tail}$, we sample a node $v_{sample}$ from the multimodal distribution $\text{Multi}(\boldsymbol{s}_{tail})$, where $\boldsymbol{s}_{tail}$ is the row vector of $S$ corresponding to $v_{tail}$. Then we construct a new similarity-aware neighborhood for $v_{tail}$ by interpolating between neighbor distributions of $v_{tail}$ and $v_{target}$. Here, the neighbor distribution for node $v$ is defined as $p(u|v) = 1/|\mathcal{N}(v)|$ if $u \in \mathcal{N}(v)$ and $p(u|v) = 0$ otherwise. To avoid the detrimental connectivity, the similarity $\text{sim}(\boldsymbol{h}_{tail}, \boldsymbol{h}_{sample})$ is used as the interpolation ratio $\phi$,

$$p_{sample}(u|v_{tail}) = (1 - \phi)p(u|v_{tail}) + \phi p(u|v_{sample}). \tag{7}$$

The interpolation ratio $\phi$ decreases as the similarity between $v_{tail}$ and $v_{sample}$ decreases, and we guarantee $(1 - \phi)$ to be at least 0.5 to preserve the original neighborhood. Given the neighbor distribution $p_{sample}(u|v_{tail})$, we sample neighbors from it without replacement. The number of neighbors is sampled from degree distribution except tail nodes to keep degree statistics.

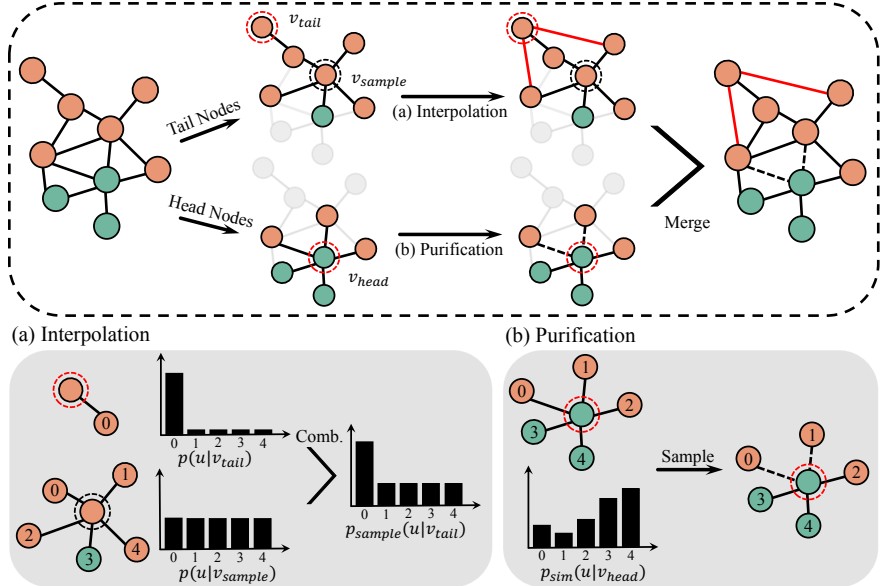

Figure 2: Topology augmentation in GRADE. Different augmentation strategies are designed for tail nodes and head nodes. Tail nodes obtain more intra-community edges by interpolation, while head nodes remove inter-community edges via purification.

For each head node $v_{head}$, we define the similarity distribution for purification. Specifically, the similarity distribution for node $v$ is $p_{sim}(u|v) = sim(\boldsymbol{h}_u, \boldsymbol{h}_v)$ if $u \in \mathcal{N}(v)$ and $p(u|v) = 0$ otherwise. Based on the similarity distribution $p_{sim}(u|v_{head})$, we sample $d_{head}(1 - p_{edr})$ neighbors without replacement, where $p_{edr}$ is the edge drop rate. Through this sampling, edges of dissimilar nodes tend to be removed, thereby retaining effective neighborhood information.

**Feature Augmentation** We randomly sample a mask vector $\boldsymbol{m} \in \{0,1\}^B$ to hide a fraction of dimensions in node feature. Each element in mask $\boldsymbol{m}$ is sampled from a Bernoulli distribution $\mathrm{Ber}(1 - p_{fdr})$, where the hyperparameter $p_{fdr}$ is the feature drop rate. Thus, the augmented node feature $\hat{X}$ is computed by

$$\hat{X} = [\boldsymbol{x}_1 \circ \boldsymbol{m}, \boldsymbol{x}_2 \circ \boldsymbol{m}, \cdots, \boldsymbol{x}_N \circ \boldsymbol{m}]. \tag{8}$$

In our implementation, we set a threshold $\zeta$ to distinguish tail nodes and head nodes. The same hyperparameters $p_{fdr}$ and $p_{edr}$ are used to generate augmentations $\hat{G}_1$ and $\hat{G}_2$.

## 4.2 Optimization Objective

We employ a contrastive objective [38] on obtained node representations of two graph augmentations. For node $v_i$, node representations $\boldsymbol{h}_i$ and $\boldsymbol{o}_i$ from different graph augmentations form the positive pair, and node representations of other nodes in two graph augmentations are regarded as negative pairs. Therefore, we define the pairwise objective for each positive pair $(\boldsymbol{h}_i, \boldsymbol{o}_i)$ as

$$\ell(\boldsymbol{h}_i, \boldsymbol{o}_i) = \log \frac{e^{\theta(\boldsymbol{h}_i, \boldsymbol{o}_i)/\tau}}{e^{\theta(\boldsymbol{h}_i, \boldsymbol{o}_i)/\tau} + \sum_{k \neq i} e^{\theta(\boldsymbol{h}_i, \boldsymbol{o}_k)/\tau} + \sum_{k \neq i} e^{\theta(\boldsymbol{h}_i, \boldsymbol{h}_k)/\tau}}. \tag{9}$$

where $\tau$ is a temperature parameter. The critic $\theta(\boldsymbol{h}, \boldsymbol{o})$ is defined as $sim(g(\boldsymbol{h}), g(\boldsymbol{o}))$, where the projection $g$ is a two-layer multilayer perceptron (MLP) to enhance the expression power [5]. The overall objective to be maximized is the average of all positive pairs,

$$\mathcal{J} = \frac{1}{2N} \sum_{i=1}^{N} \left[ \ell(\boldsymbol{h}_i, \boldsymbol{o}_i) + \ell(\boldsymbol{o}_i, \boldsymbol{h}_i) \right]. \tag{10}$$

Table 1: Quantitative results (%) on node classification. (bold: best; em dash: out-of-memory)

| | | Cora | | Citeseer | | Photo | | Computer | |
|---|---|---|---|---|---|---|---|---|---|
| | | *Micro-F1* | *Macro-F1* | *Micro-F1* | *Macro-F1* | *Micro-F1* | *Macro-F1* | *Micro-F1* | *Macro-F1* |
| **Supervised Split** | **GCN** | $82.30_{\pm0.49}$ | $76.87_{\pm0.34}$ | $65.84_{\pm0.55}$ | $59.62_{\pm0.64}$ | $93.52_{\pm0.82}$ | $\mathbf{78.88_{\pm2.01}}$ | $89.14_{\pm0.75}$ | $72.61_{\pm3.05}$ |
| | **DGI** | $82.28_{\pm0.84}$ | $77.23_{\pm0.90}$ | $65.64_{\pm0.63}$ | $59.47_{\pm1.24}$ | $92.98_{\pm1.12}$ | $78.83_{\pm1.66}$ | $88.96_{\pm0.96}$ | $72.30_{\pm1.80}$ |
| | **GraphCL** | $81.78_{\pm0.67}$ | $76.01_{\pm1.07}$ | $65.16_{\pm1.02}$ | $58.72_{\pm1.37}$ | — | — | — | — |
| | **GRACE** | $82.32_{\pm0.45}$ | $76.78_{\pm0.87}$ | $64.16_{\pm2.07}$ | $59.73_{\pm1.94}$ | $93.12_{\pm0.40}$ | $78.60_{\pm3.12}$ | $88.22_{\pm1.04}$ | $71.74_{\pm3.05}$ |
| | **MVGRL** | $83.22_{\pm1.02}$ | $77.84_{\pm1.35}$ | $66.26_{\pm0.72}$ | $60.30_{\pm0.95}$ | $94.10_{\pm0.31}$ | $78.36_{\pm2.22}$ | — | — |
| | **CCA-SSG** | $82.70_{\pm0.86}$ | $77.35_{\pm1.06}$ | $65.96_{\pm1.36}$ | $58.81_{\pm1.67}$ | $94.36_{\pm0.25}$ | $79.34_{\pm3.42}$ | $89.22_{\pm0.95}$ | $73.82_{\pm1.80}$ |
| | **GRADE** | $\mathbf{83.40_{\pm0.80}}$ | $\mathbf{78.54_{\pm1.15}}$ | $\mathbf{67.14_{\pm1.07}}$ | $\mathbf{61.04_{\pm2.07}}$ | $\mathbf{94.72_{\pm0.30}}$ | $78.86_{\pm2.77}$ | $\mathbf{89.42_{\pm0.53}}$ | $\mathbf{74.71_{\pm1.30}}$ |
| **Semi-supervised Split** | **GCN** | $74.18_{\pm0.40}$ | $69.84_{\pm0.56}$ | $53.80_{\pm0.94}$ | $50.15_{\pm0.69}$ | $91.04_{\pm0.65}$ | $65.47_{\pm1.20}$ | $78.58_{\pm0.93}$ | $61.80_{\pm1.43}$ |
| | **DGI** | $75.92_{\pm0.86}$ | $70.04_{\pm0.53}$ | $54.52_{\pm1.44}$ | $51.92_{\pm1.23}$ | $90.78_{\pm0.78}$ | $66.27_{\pm0.76}$ | $79.00_{\pm0.80}$ | $62.00_{\pm1.70}$ |
| | **GraphCL** | $75.68_{\pm2.84}$ | $69.86_{\pm2.41}$ | $54.06_{\pm1.93}$ | $51.75_{\pm1.78}$ | — | — | — | — |
| | **GRACE** | $75.12_{\pm1.41}$ | $69.66_{\pm1.29}$ | $53.56_{\pm3.42}$ | $49.83_{\pm1.74}$ | $91.12_{\pm0.31}$ | $65.07_{\pm1.28}$ | $79.10_{\pm1.79}$ | $61.76_{\pm1.97}$ |
| | **MVGRL** | $76.44_{\pm1.17}$ | $70.52_{\pm1.63}$ | $56.84_{\pm1.26}$ | $53.79_{\pm1.25}$ | $92.01_{\pm0.87}$ | $66.16_{\pm2.13}$ | — | — |
| | **CCA-SSG** | $75.74_{\pm1.96}$ | $71.70_{\pm1.59}$ | $57.90_{\pm1.82}$ | $54.70_{\pm1.54}$ | $91.68_{\pm0.50}$ | $\mathbf{67.08_{\pm1.08}}$ | $82.20_{\pm0.47}$ | $65.04_{\pm1.16}$ |
| | **GRADE** | $\mathbf{77.20_{\pm0.94}}$ | $\mathbf{73.37_{\pm1.27}}$ | $\mathbf{59.44_{\pm0.78}}$ | $\mathbf{56.47_{\pm0.64}}$ | $\mathbf{92.04_{\pm0.30}}$ | $66.62_{\pm2.27}$ | $\mathbf{82.50_{\pm1.04}}$ | $\mathbf{67.50_{\pm1.80}}$ |

## 5 Experiments

**Datasets** For a comprehensive comparison, we use four real-world datasets to evaluate the performance of node classification and the fairness to degree bias. Specifically, we choose two categories of datasets: 1) citation networks including Cora [17] and Citeseer [17], 2) social networks Photo [23] and Computer [23] from Amazon. The statistics of these datasets are summarized in Appendix C.

**Evaluation Protocol** We compare GRADE with state-of-the-art GCL models DGI [30], GraphCL [35], GRACE [38], MVGRL [12] and CCA-SSG [36], and semi-supervised baseline GCN [17] for reference. For GCL models, we follow the linear evaluation scheme introduced in [30], where each model is firstly trained in an unsupervised manner and node representations are subsequently fed into a simple logistic regression classifier. We adopt two universally accepted splits for full evaluation: 1) semi-supervised split [30, 35] that 20 labeled nodes per class are for training and 1000 nodes are for testing, 2) supervised split [38, 36] that 1000 nodes are for testing and the rest of nodes form the training set. It is worth noting that 1000 nodes in the test set are randomly sampled with degrees less than 50 to provide an appropriate degree range for analysis. GCN is trained by the original paradigm [17] with the above train-test split. We refer readers of interest to Appendix C on details of experiments, including implementation and hyperparameters.

### 5.1 Main Results and Analysis

**Node Classification** We train each model for 10 independent trials with different seeds, and report mean and standard deviation results in Table 1. We observe that the proposed GRADE outperforms all baselines in most cases. The improvement of GRADE is more pronounced on Cora and Citeseer datasets, where average node degrees are around 3 and a large number of tail nodes exist. Additionally, we find that GCL models tend to outperform GCN in the semi-supervised split, suggesting that GCN may benefit more from end-to-end training with more supervision.

To verify that GRADE improves the classification performance of tail nodes and also retains the performance of head nodes, we divide test nodes of Cora into tail nodes and head nodes based on the threshold $\zeta$. We draw average accuracy w.r.t. degree of GRADE and competitive baselines as violin plots in Figure 3. The subsequent experiments default to the supervised split if not specified. As expected, GRADE achieves obvious performance gain regardless of tail nodes or head nodes.

**Fairness Analysis** In order to quantitatively analyze the fairness to degree bias, we define the group mean as the mean of degree-specific average accuracy while the bias is the variance. Mathematically,

$$\text{Avg. Acc.}(k) = \mathbb{E}[\{\text{Acc}(v_i), \forall \text{ node } v_i \text{ such that } d_i = k\}],$$
$$G.Mean = \mathbb{E}[\{\text{Avg. Acc.}(k), \forall \text{ node degree } k\}], Bias = \text{Var}(\{\text{Avg. Acc.}(k), \forall \text{ node degree } k\}).$$

Table 2: Quantitative results (%) on fairness analysis.

| | Cora | | Citeseer | | Photo | | Computer | |
|---|---|---|---|---|---|---|---|---|
| | *G. Mean↑* | *Bias↓* | *G. Mean↑* | *Bias↓* | *G. Mean↑* | *Bias↓* | *G. Mean↑* | *Bias↓* |
| **GCN** | 86.04 | 1.70 | 84.00 | 1.85 | 97.41 | 0.28 | 96.30 | 0.50 |
| **DGI** | 89.26 | 0.67 | 84.79 | 1.71 | 98.23 | 0.27 | 96.94 | 0.45 |
| **GraphCL** | 90.80 | 0.59 | 84.13 | 1.80 | — | — | — | — |
| **GRACE** | 89.91 | 0.70 | 85.44 | 1.67 | 98.28 | 0.23 | 96.92 | 0.47 |
| **MVGRL** | 91.01 | 0.54 | 83.86 | 1.83 | 98.39 | 0.27 | — | — |
| **CCA-SSG** | 90.86 | 0.63 | 84.35 | 1.73 | 98.44 | 0.24 | 97.17 | 0.39 |
| **GRADE** | **92.87** | **0.48** | **85.88** | **1.52** | **98.52** | **0.20** | **97.42** | **0.35** |

Based on these metrics, evaluation results are shown in Table 2. It can be seen that GRADE reduces the bias across all datasets and maintain the highest group mean. Moreover, graph contrasting learning models have a smaller bias compared to GCN, conforming to our previous study in Section 2.

**Visualization**  To demonstrate that GRADE pulls same-community node representations more concentrated, we visualize node representations of GRADE and competitive baselines on the Cora dataset in Figure 5. Particularly, we zoom into one specific community colored blue. Graph contrastive learning baselines always depict more crisp boundaries than GCN, where blue nodes are still scattered in space. In GRADE, they are finally clustered together, illustrating that the proposed augmentation design exerts a vital part.

## 5.2 Ablation Study and Hyperparameter Sensitivity

**Ablation Study of Sampling**  Recall that GRADE augments tail nodes by sampling nodes for interpolation, and also sample edges to remove in the purification of head nodes. We alter the augmentation by fixing to the most similar node (without random interpolation) or top-$d_{head}(1-p_{edr})$ similar nodes (without random deleting) to validate the effectiveness of these sampling processes. Results of the ablation study on Cora and Citeseer datasets are reported in Table 3. We can observe that GRADE is consistently better than the remaining variants. Such a phenomenon implies that reasonable randomness provides more diverse node contexts to contrast out essential features.

**Effect of Threshold**  We investigate the impact of threshold $\zeta$ used to split tail nodes and head nodes on classification performance. Figure 4 (a) shows the test Micro-F1 w.r.t. different $\zeta$ on Cora dataset. The performance benefits from an applicable selection of $\zeta$. When $\zeta$ is too small, the neighbor sparsity of tail nodes cannot be mitigated; if it is too large, noise is injected.

**Effect of Drop Rate**  We perform sensitivity analysis on feature drop rate $d_{fdr}$ and edge drop rate $d_{edr}$ which control the generation of graph augmentations. We vary these hyperparameters from 0 to 0.5 in node classification on the Cora dataset. The results are shown in Figure 4 (b). It can be observed that the performance is relatively poor when the hyperparameter $d_{fdr}$ is too large and $d_{edr}$ is too small. We infer that if the $d_{fdr}$ is too large, the original graph is heavily undermined to contain useful information; and if the $d_{edr}$ is too small, the large number of neighbors makes the perturbation of different augmentations too insignificant to achieve the purpose of contrast.

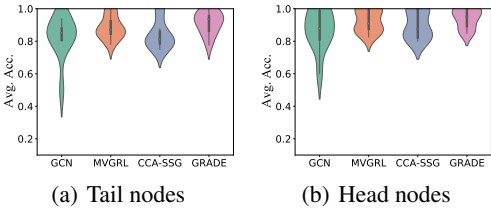

(a) Tail nodes        (b) Head nodes

Figure 3: Violin plots of the average accuracy w.r.t. node degree for (a) tail nodes and (b) head nodes on the Cora dataset. The box inside the violin indicates 25-75 percentiles, and the median is shown by a white scatter.

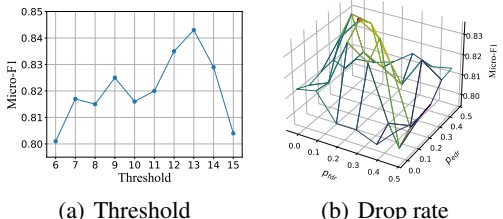

(a) Threshold        (b) Drop rate

Figure 4:  The hyperparameter sensitivity of GRADE with varying (a) threshold and (b) drop rate on Cora dataset. In (b), the lighter color represents better performance, and the red scatter highlights the peak.

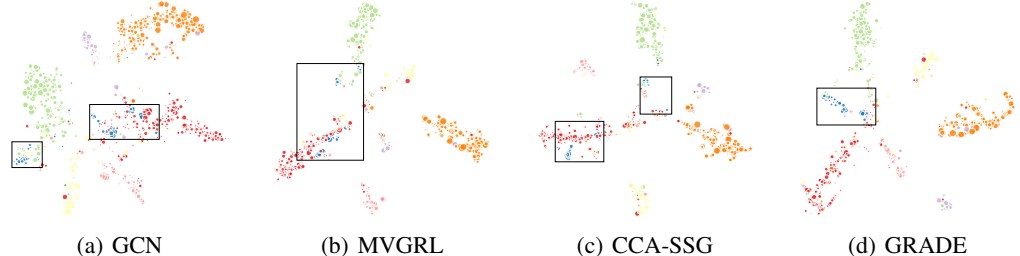

| (a) GCN | (b) MVGRL | (c) CCA-SSG | (d) GRADE |

Figure 5: Visualization of node representations learned by competitive baselines and GRADE on Cora dataset. Color denotes the community of nodes and size represents the node degree. Black boxes highlight one community colored by blue as an example.

Table 3: Ablation study (%) on the sampling of GRADE. (w/o RI: without random interpolation; w/o RD: without random deleting)

| | Cora | | | | Citeseer | | | |
|---|---|---|---|---|---|---|---|---|
| | *Micro-F1*↑ | *Macro-F1*↑ | *G. Mean*↑ | *Bias*↓ | *Micro-F1*↑ | *Macro-F1*↑ | *G. Mean*↑ | *Bias*↓ |
| **w/o RI** | 83.00 | 77.97 | 90.86 | 0.55 | 66.10 | 61.02 | 84.96 | 1.80 |
| **w/o RD** | 82.30 | 76.50 | 90.54 | 0.60 | 65.90 | 58.83 | 84.21 | 1.84 |
| **w/o RI+RD** | 82.00 | 75.82 | 89.97 | 0.63 | 65.80 | 59.29 | 83.00 | 1.95 |
| **GRADE** | **84.30** | **80.66** | **92.87** | **0.48** | **68.80** | **64.21** | **85.88** | **1.52** |

# 6 Related Work

**Graph Neural Networks**   Graph neural networks (GNNs) [18, 32, 4, 31] can be generally divided into spectral methods and spatial methods. Specifically, spectral methods learn node representations based on graph spectral theory. [3] first proposes a spectral graph-based extension of convolutional networks, and GCN [17] simplifies ChebNet [6] by the first-order approximation. Spatial methods directly define graph convolution in the spatial domain. GraphSAGE [11] learns aggregators by sampling and aggregating neighbors, and GAT [29] assigns different edge weights during aggregation. We refer readers to recent surveys [34, 37] for a more comprehensive review. On the other hand, some literature [25, 33, 21] shows that there is a structural unfairness between tail nodes and head nodes in GNNs. Existing explorations [20, 16] focus on supervised settings, ignoring the great potential of self-supervised learning on this problem.

**Graph Contrastive Learning**   Being popular in self-supervised visual representation learning [28, 7, 27, 13, 5, 10, 15], contrastive learning obtains discriminative representations by contrasting positive and negative samples. Inspired by the local-global mutual information maximization viewpoint [14], DGI [30] and InfoGraph [24] first marry the power of GNNs and contrastive learning. Following them, MVGRL [12] introduces the node diffusion to the graph contrastive framework. GRACE [38], GCA [39] and GraphCL [35] learn node representations by treating other nodes as negative samples, while BGRL [26] proposes a negative-sample-free model. CCA-SSG [36] optimizes a feature-level objective other than instance-level discrimination. There are several surveys [1, 19] summarizing recent advances in graph contrastive learning. Despite their remarkable achievements, there is no graph contrastive learning targeting the fairness of degree bias.

# 7 Conclusion

In this paper, we bring to light the prospect of GCL to alleviate structural unfairness for node representation learning. We discover that node representations obtained by GCL methods are fairer to degree bias than those learned by GCN, and explore the underlying cause of this phenomenon. Based on our theoretical analysis, we further propose a novel GCL model targeting degree bias.

**Limitations and Broader Impact**   A limitation of GRADE is its heuristic design, therefore an interesting direction for future work is to extend GRADE into learnable graph augmentation. Our work investigates the structural fairness in GCL for the first time and points out the great potential of GCL for this problem. Considering that most real-world graphs follow the long-tail distribution, the studied problem is practical and important. Additionally, our work deepens the understanding of the learning mechanism of GCL, and may inspire more future research on structural fairness.

