# OpenReview forum: "Uncovering the Structural Fairness in Graph Contrastive Learning"
_NeurIPS.cc/2022/Conference — NeurIPS 2022 Accept_

### Official Review · Reviewer_x4f1 · 2022-07-09

**Rating:** 7
**Confidence:** 5
**Soundness:** 2 fair
**Presentation:** 3 good
**Contribution:** 3 good

**Summary:**

Node degrees of real-world graphs follow a long-tailed distribution, but GCN exhibits a performance disparity between high-degree nodes and low-degree nodes, i.e., degree bias. This paper discovers an interesting phenomenon that graph contrastive learning methods have already a smaller degree bias.  Based on this discovery, this paper theoretically analyzes the reason and proposes a tailored contrastive learning method GRADE. Experiments validate the effectiveness of the proposed method.


**Questions:**

1. Is the conclusion only suitable for GCN? I wonder GAT may exhibit a smaller degree bias, even smaller than graph contrastive learning methods.
2. From Figure 6 in Appendix A, the advantage of graph contrastive learning methods over GCN on Photo dataset is not obvious.
3. How does a clear community structure lead to a smaller degree bias?
4. The improvement of the proposed method in Table 1 does not seem statistically significant because of high variance. And the related works designed for degree bias are not set as baselines in the experimental comparison.


**Limitations:**

In addition to the limitations mentioned in the paper, the generalization of the conclusion should be taken into consideration.

**Strengths And Weaknesses:**

Strength:
1.	The finding is interesting and may inspire a new paradigm for alleviating degree bias.
2.	The paper is well-written and the conclusion is clear.
Weakness:
1.	The conclusion seems to be only for GCN. I wonder GAT[1] may exhibit a smaller degree bias, even smaller than graph contrastive learning methods.
2.	From Figure 6 in Appendix A, the advantage of graph contrastive learning methods over GCN on Photo dataset is not obvious. The numerical values of their slopes are close.
3.	There is a small gap between degree bias and theoretical analysis of clear community structure.
4.	The improvement of the proposed method in Table 1 does not seem statistically significant because of high variance.
5.	There are some related works designed for degree bias, such as SL-DSGCN[2]. But these methods are not set as baselines in the experimental comparison.

[1] Veličković P, Cucurull G, Casanova A, et al. Graph Attention Networks[C]//International Conference on Learning Representations. 2018.
[2] Tang X, Yao H, Sun Y, et al. Investigating and mitigating degree-related biases in graph convoltuional networks[C]//Proceedings of the 29th ACM International Conference on Information & Knowledge Management. 2020: 1435-1444.

---

> ### Author Response · Authors · 2022-08-02
> **Response**
>
> - We highly appreciate constructive comments from the Reviewer on our work.
>
> 1. > Is the conclusion only suitable for GCN?
>
> **Answer:** We only focus on GCN because almost all graph contrastive learning methods default to GCN as the encoder. In other words, we fix the encoders as GCN, and explore the advantages of the contrastive learning framework compared to the semi-supervised framework w.r.t. degree bias.
>
> 2. > From Figure 6 in Appendix A, the advantage of graph contrastive learning methods over GCN on Photo dataset is not obvious.
>
> **Answer:**  Thanks for your careful reading. Compared to Cora and Citeseer, the gap between slopes of graph contrastive learning methods and GCN is relatively small on Photo. A reasonable hypothesis is that the average node degree of Photo dataset is much larger than those of Cora and Citeseer datasets, where the advantage of GCL to alleviate the neighborhood sparsity of tail nodes cannot be well exhibited.
>
> 3. > How does a clear community structure lead to a smaller degree bias?
>
> **Answer:** We are sorry for the unclear statement. Intuitively, a clear community structure makes representations of tail nodes closer to those in the same community. These refined representations drive tail nodes away from the community boundary, thus tail nodes obtain better classification performance. Experimentally, visualization in Figure 5 and fairness analysis in Table 2 further demonstrate that a clearer community structure does alleviate the degree bias. We will add these explanations to our paper in the revision. Thanks.
>
> 4. > The improvement of the proposed method in Table 1 does not seem statistically significant because of high variance. SL-DSGCN designed for degree bias is not set as baselines in the experimental comparison.
>
> **Answer:** Thanks for your suggestions. (1) We calculate confidence intervals of the difference distribution between GRADE and baselines in Table 1, and obtain the following statistical results: in the supervised split, GRADE outperforms all baselines on Citeseer, 5 out of 6 on Cora, Photo and Computer with over 95% confidence; in the semi-supervised split, GRADE outperforms all baselines on Cora and Citeseer, 4 out of 6 on Photo and 5 out of 6 on Computer with over 95% confidence. The improvement of GRADE is more pronounced on Cora and Citeseer, because their average node degrees are around 3 and a large number of tail nodes exist. Therefore, it’s safe to say that GRADE outperforms most baselines with statistical significance. (2) SL-DSGCN is a semi-supervised GNN, while our work focuses on unsupervised graph contrastive learning. Therefore, they are not comparable.

---

### Official Review · Reviewer_VyUL · 2022-07-10

**Rating:** 7
**Confidence:** 4
**Soundness:** 3 good
**Presentation:** 3 good
**Contribution:** 3 good

**Summary:**

This paper investigates the great potential of graph contrastive learning to solve the degree-bias problem, and proposes a new graph augmentation for further improvement.

**Questions:**

1. Do other semi-supervised GNNs suffer from the same severe degree-bias problem?
2. Some simplifications have been made in the theoretical analysis. For example, they only consider topology augmentation. Does it affect the analysis results?
3. There is only one train-test split in Section 2, but two in Section 5.
4. The complexity of the proposed model.


**Limitations:**

The authors have pointed out the limitations of their work.

**Strengths And Weaknesses:**

The motivation of this paper is clear. They discover that node representations obtained by GCL methods are fairer to degree bias than those learned by GCN, and explore the underlying cause of this phenomenon. Based on the theoretical analysis, they further propose a novel GCL model targeting the degree bias. Experimental results clearly show the merit of the proposed model, and the source code is attached (I have not run it though).

Concerns:
1. This paper focuses on semi-supervised GCN to motivate their investigation. Do other semi-supervised GNNs suffer from the same severe degree-bias problem? In other words, I doubt the finding is limited.
2. Some simplifications have been made in the theoretical analysis. For example, they only consider the topology augmentation. Does it affect the analysis results?
3. There is only one train-test split in Section 2, but two in Section 5.
4. The complexity of the proposed model.

---

> ### Author Response · Authors · 2022-08-02
> **Response**
>
> - We sincerely thank the Reviewer for spending time and providing valuable feedback. We appreciate all of your suggestions and we have addressed all your questions below by providing our responses.
>
> 1. > Do other semi-supervised GNNs suffer from the same severe degree-bias problem?
>
> **Answer:** Whether other semi-supervised GNNs suffer from the same severe degree-bias is not the focus of our work. We only focus on GCN because almost all graph contrastive learning methods default to GCN as the encoder. In other words, we fix the encoders as GCN, and explore the advantages of the contrastive learning framework compared to the semi-supervised framework w.r.t. degree bias.
>
> 2. > They only consider topology augmentation. Does it affect the analysis results?
>
> **Answer:** We focus on topology augmentation not only to simplify theoretical analysis, but also because it is unique to graph augmentation. Feature augmentation is independent of topology augmentation strategy, and theoretical analysis in machine learning [1] has proved that it also has the effect of representation concentration.
>
> 3. > There is only one train-test split in Section 2, but two in Section 5.
>
> **Answer:** Thank you for your careful reading. Both the original papers of DGI and GraphCL in Section 2 employ the semi-supervised split, so we follow their experimental settings for comparison. However, the baselines in Section 5, such as CCA-SSG [2] and GRACE [3], use supervised split. For a full comparison, we conduct experiments under both experimental settings.
>
> 4. > The complexity of the proposed model.
>
> **Answer:** Thank you for your suggestion. We now conduct a complexity analysis. The overhead of GRADE mainly lies in the neighborhood interpolation and neighbor sampling. The time complexity of calculating the cosine similarity and neighbor distribution is $O(N^2D + N^2)$, where $D$ is the representation dimension. And the neighbor sampling costs $O(N)$. Considering that $D \ll N$, the overall time complexity is around $O(N^2)$. Additionally, we implement the model based on PyTorch, and the tensor operation can be parallelized on GPUs. Therefore, the actual overhead is smaller than $O(N^2)$, indicating that the proposed GRADE is still efficient.
>
> [1] Weiran Huang, Mingyang Yi, and Xuyang Zhao. Towards the generalization of contrastive self-supervised learning. *arXiv e-prints*, 2021.
>
> [2] Hengrui Zhang, Qitian Wu, Junchi Yan, David Wipf, and Philip S Yu. From canonical correlation analysis to self-supervised graph neural networks. In *NeurIPS*, 2021.
>
> [3] Yanqiao Zhu, Yichen Xu, Feng Yu, Qiang Liu, Shu Wu, and Liang Wang. Graph contrastive learning with adaptive augmentation. In *WWW*, 2021.

---

### Official Review · Reviewer_va2A · 2022-07-11

**Rating:** 8
**Confidence:** 4
**Soundness:** 3 good
**Presentation:** 3 good
**Contribution:** 3 good

**Summary:**

GCN is primarily beneficial to high-degree nodes but biased against low-degree nodes, which causes a performance bottleneck. As a promising paradigm in the graph domain, GCL integrates the power of GCN and contrastive learning, displaying SOTA performance in a variety of tasks. This paper investigates the question of whether will GCL present the same degree of bias as GCN. They surprisingly find out that a smaller performance gap exists between tail nodes and head nodes in GCL methods than that of GCN.

They intuitively and theoretically analyze the reason for this interesting finding. Particularly, Intra-community Concentration Theorem and Inter-community Scatter Theorem prove that node representations learned by GCL conform to a clearer community structure, and establish the relation between graph augmentation and representation concentration. These analyses yield profound insights into solutions to this important degree-bias problem and imply that GCL is a promising direction.

Therefore, they further propose a GRAph contrastive learning for DEgree bias (GRADE) to concentrate augmented representations. Specifically, they enlarge limited neighbors of tail nodes to contain more nodes within the same community and purify head nodes by removing neighbors from a different community. Extensive experiments on various benchmark datasets and several evaluation protocols validate the effectiveness of GRADE.


**Questions:**

Q1: How to get a small degree of bias from a clear community structure?
Q2: Why is the supremum in Definition 1 \gamma(\frac{B}{\hat{d}_{\min}^k})^{\frac{1}{2}}? Based on this definition, how to prove that the proposed GRADE reduces this supremum?
Q3: There is a lack of significant test results in Table 1.



**Limitations:**

A learnable augmentation is an interesting direction for improvement.

**Strengths And Weaknesses:**

The paper is well-written in general and their finding is exciting.
Weaknesses:
1.	How to get a small degree of bias from a clear community structure needs more explanations. Theorem 1 and 2 prove that GCL conforms to a clearer community structure via intra-community concentration and inter-community scatter, but its relationship with degree bias is not intuitive enough.
2.	There is some confusion in the theoretical analysis. Why is the supremum in Definition 1 \gamma(\frac{B}{\hat{d}_{\min}^k})^{\frac{1}{2}}? Based on this definition, how to prove that the proposed GRADE reduces this supremum?
3.	There is a lack of significance test in Table 1.
Despite the weaknesses mentioned above, I believe that this paper is worth publishing. They consider an important degree-bias problem in the graph domain, given that node degrees of real-world graphs often follow a long-tailed power-law distribution. And they show an exciting finding that GCL is more stable w.r.t. the degree bias, and give a preliminary explanation for the underlying mechanism. Although the improvement does not seem significant in Table 1, they may inspire more future research on this promising solution.

---

> ### Author Response · Authors · 2022-08-02
> **Response**
>
> - We sincerely thank the Reviewer for all the comments, and it is a great honor for us to inspire your interest. We have addressed all your questions below and hope they have clarified all confusion you had about our work.
>
> 1. > How to get a small degree of bias from a clear community structure?
>
> **Answer:** We are sorry for the unclear statement, which causes your confusion. Intuitively, a clear community structure makes representations of tail nodes closer to those in the same community. These refined representations drive tail nodes away from the community boundary, thus tail nodes obtain better classification performance for narrowing the bias. Experimentally, visualization in Figure 5 and fairness analysis in Table 2 further demonstrate that a clearer community structure does alleviate the degree bias. Thank you for your careful reading. We will revise the relevant parts carefully in the revision.
>
> 2. > Why is the supremum in Definition 1 $\gamma(\frac{B}{\hat{d}_{\min}^k})^{\frac{1}{2}}$? Based on this definition, how to prove that the proposed GRADE reduces this supremum?
>
> **Answer:**  (1) Actually, we derive the supremum $\gamma(\frac{B}{\hat{d}_{\min}^k})^{\frac{1}{2}}$ in Appendix B. In a nutshell,  there is a natural supremum
>
> $$2\sqrt{B}\cdot\min_{\hat{\mathcal{G}}_i \in \mathcal{T}(\mathcal{G}_i), \hat{\mathcal{G}}_j \in \mathcal{T}(\mathcal{G}_j)}\frac{1}{\sqrt{\hat{d}_j}}$$
>
> for $d_{\mathcal{T}}(v_i, v_j)$ . Following this form, we define the supremum
>
> $$\gamma(\frac{B}{\hat{d}_{\min}^k})^{\frac{1}{2}}$$
>
> to delineate the concentrated part. (2) The ($\alpha,\gamma,\hat{d}$)-Augmentation is defined as a measure of representation concentration, not an optimization goal. Given $\alpha$, smaller $\gamma(B/\hat{d}_{\min}^k)^{\frac{1}{2}}$ indicate representations are more concentrated and vice versa. In experiments, we intuitively validate that GRADE makes node representations more concentrated in the visualization of Figure 5.
>
> 3. > There is a lack of significant test results in Table 1.
>
> **Answer:** Thanks for the suggestion. We calculate confidence intervals of the difference distribution between GRADE and baselines in Table 1, and obtain the following statistical results: in the supervised split, GRADE outperforms all baselines on Citeseer, 5 out of 6 on Cora, Photo and Computer with over 95% confidence; in the semi-supervised split, GRADE outperforms all baselines on Cora and Citeseer, 4 out of 6 on Photo and 5 out of 6 on Computer with over 95% confidence. The improvement of GRADE is more pronounced on Cora and Citeseer, because their average node degrees are around 3 and a large number of tail nodes exist. Therefore, it’s safe to say that GRADE outperforms most baselines with statistical significance.

---

> > ### Comment · Reviewer_va2A · 2022-08-07
> > **Reply to author's response**
> >
> > Thank you for your detailed response. Overall, I think this work is solid and interesting. I vote for an acceptance.

---

### Official Review · Reviewer_5815 · 2022-07-12

**Rating:** 7
**Confidence:** 4
**Soundness:** 3 good
**Presentation:** 3 good
**Contribution:** 3 good

**Summary:**

This paper studies structural fairness on graph contrastive learning (GCL). The study is motivated by the finding that GCL is fairer to low degree nodes than GCN. Based on that, the authors first present theoretical analysis on such structural fairness for GCL through intra-community concentration theorem and inter-community scatter theorem. Guided by the theoretical analysis, the authors propose GRADE by enriching the neighborhood of tail nodes and purifying the neighborhood of head nodes. Experimental results on real-world datasets demonstrate the effectiveness of GRADE.

**Questions:**

- How correlated between the community assignment and the node labels? If there are high correlation between them, enhancing intra-community edges and purifying inter-community edges is an intuitive choice to debias.

- Related to the previous questions, does the theoretical analysis implicitly assume the graph to be of high homophily? What would be the performance on graphs with heterophily?

- For GCN in this paper, does it use the random walk graph Laplacian or the commonly used symmetrically normalized graph Laplacian?

- How would the theoretical results change if we use the symmetric normalized graph Laplacian (as in original GCN) rather than random walk graph Laplacian?

- What is the \kappa in Theorem 1?

- (Just some random thoughts) Rather than bringing high-level idea (enhancing intra-community edges and purifying inter-community edges) to methodology design, can we have more insights from the theoretical analysis? For example, is it possible that the bounds for intra-community concentration and inter-community scatter could help us identify what a good augmentation should look like? Then we can sample/augment edges guided by these bounds.


**Limitations:**

- More understanding on the performance for graphs with heterophily is needed.

- There are too many notations in the paper. The authors may want to find a way to organize them so the readers won't need to check notations back and forth.

- The authors should make sure the theorems are self-contained.

- Significance test on the experimental results would be helpful to showcase the effectiveness of GRADE.

**Strengths And Weaknesses:**

Strengths:
- The paper is well-motivated through empirical experiments.
- The design of the proposed method is inspired from theoretical analysis.
- The proposed method is effective in mitigating structural unfairness as shown by experimental results.

Weaknesses:
Please see below.

---

> ### Author Response · Authors · 2022-08-02
> **Response**
>
> - We sincerely thank the Reviewer for your careful reading and thought-provoking insights. We are sorry for the negligence in the description of the theorem, and will correct them in our paper. We would like to address the concerns of the Reviewer by providing responses as well as additional experimental results.
>
> 1. > How correlated between the community assignment and the node labels? If there are high correlation between them, enhancing intra-community edges and purifying inter-community edges is an intuitive choice to debias.
>
> **Answer:** Since the homophily of graphs usually holds, community assignment and node labels are generally correlated. In this case, GRADE enhances intra-community edges and purifies inter-community edges to debias downstream tasks.
>
> 2. > Related to the previous questions, does the theoretical analysis implicitly assume the graph to be of high homophily? What would be the performance on graphs with heterophily?
>
> **Answer:** Thanks for your comments and this is a very good question. In fact, we agree that the theoretical analysis implicitly assumes the graph to be of high homophily. Unsupervised graph contrastive learning is difficult to perform well if community assignment and node labels are uncorrelated. Likewise, GRADE is not specifically designed for heterophily and not expected to be better than methods tailed for disassortative graphs.
>
> To investigate the performance of GRADE and baselines on disassortative graphs, we conduct node classification on disassortative benchmark Chameleon and report the results in Table 1 below.
>
> Table 1	Quantitative results (%) on node classification and fairness analysis in semi-supervised split. (bold: best)
>
> | Metrics             | GCN   | DGI      | GraphCL | GRACE | MVGRL | CCA-SSG | GRADE     |
> | ------------------- | ----- | -------- | ------- | ----- | ----- | ------- | --------- |
> | Micro-F1 $\uparrow$ | 34.41 | 30.42    | 29.11   | 35.02 | 33.04 | 37.18   | **38.48** |
> | Macro-F1 $\uparrow$ | 27.49 | 26.93    | 26.00   | 27.93 | 29.35 | 32.34   | **33.57** |
> | G. Mean $\uparrow$  | 42.25 | 33.17    | 31.62   | 39.51 | 32.44 | 41.68   | **43.01** |
> | Bias $\downarrow$   | 7.04  | **5.39** | 5.57    | 6.17  | 5.88  | 6.33    | 5.43      |
>
> Although the classification performance of all methods is not ideal on the disassortative graph, GRADE outperforms all baselines under most metrics. A reasonable hypothesis is that enhancing intra-community edges and purifying inter-community edges may improve the homophily of augmented graphs, which are beneficial to GCN encoder.
>
> 3. > For GCN in this paper, does it use the random walk graph Laplacian or the commonly used symmetrically normalized graph Laplacian? How would the theoretical results change if we use the symmetric normalized graph Laplacian (as in original GCN) rather than random walk graph Laplacian?
>
> **Answer:** (1) In our theoretical analysis, we use the random walk graph Laplacian. (2) The theoretical results remain the same if we use the symmetric normalized graph Laplacian rather than random walk graph Laplacian. Specifically, We choose random walk graph Laplacian because its row vectors are conveniently written as degrees of target nodes in augmentation distance $d_{\mathcal{T}}(v_i, v_j)$ , i.e.,
>
> $$\min_{\hat{\mathcal{G}}_i \in \mathcal{T}(\mathcal{G}_i), \hat{\mathcal{G}}_j \in \mathcal{T}(\mathcal{G}_j)}\|(\frac{\hat{A}_i}{\hat{d}_i}-\frac{\hat{A}_j}{\hat{d}_j})X\|.$$
>
> If we use symmetrically normalized graph Laplacian, the definition of  $d_{\mathcal{T}}(v_i, v_j)$ will convert to
>
> $$\min _{\hat{\mathcal{G}}_i \in \mathcal{T}(\mathcal{G}_i), \hat{\mathcal{G}}_j \in \mathcal{T}(\mathcal{G}_j), v_l \in \hat{\mathcal{G}}_i, v_m \in \hat{\mathcal{G}}_j}\|(\frac{\hat{A}_i}{\sqrt{\hat{d}_i\hat{d}_l}}-\frac{\hat{A}_j}{\sqrt{\hat{d}_j\hat{d}_m}})X\|.$$
>
> However, we can still define
>
> $$\gamma(\frac{B}{\hat{d}_{\min}^k})^{\frac{1}{2}}$$
>
> to bound $\sup_{v_i, v_j \in C_k^0} d_{\mathcal{T}}(v_i, v_j)$ in the definition of ($\alpha,\gamma,\hat{d}$)-augmentation. This means that the subsequent proofs will not be changed by symmetrically normalized graph Laplacian.

---

> > ### Author Response · Authors · 2022-08-02
> > **Response**
> >
> > 4. > What is the $\kappa$ in Theorem 1?
> >
> > **Answer:** Sorry for our unclear statement, causing your confusion. For the variance of pre-transformation representations $\sigma$ and the variance of node representations $\sigma_{f,\varepsilon}^2$, intra-community concentration means $\sigma_{f,\varepsilon}^2\leq\frac{1}{\kappa}\sigma^2$ with $\kappa \geq 1$. $\kappa$ is a coefficient that reflects the degree of concentration. Based on this coefficient $\kappa$, we derive that the condition $\varepsilon^2\leq\frac{\beta m}{6M^2\kappa}$ is required. More proofs can be found in Appendix B. We will revise it and make a clear explanation in our revision.
> >
> > 5. > (Just some random thoughts) Rather than bringing high-level idea (enhancing intra-community edges and purifying inter-community edges) to methodology design, can we have more insights from the theoretical analysis? For example, is it possible that the bounds for intra-community concentration and inter-community scatter could help us identify what a good augmentation should look like? Then we can sample/augment edges guided by these bounds.
> >
> > **Answer:** Our theoretical analysis reveals that bounds for intra-community concentration and inter-community scatter are controlled by two key factors: 1) The alignment of positive pairs. Good alignment enables small
> >
> > $$\mathbb{E}_{\hat{\mathcal{G}}_i^1,\hat{\mathcal{G}}_i^2\in\mathcal{T}(\mathcal{G}_i)}\|f(\hat{\mathcal{G}}_i^1)-f(\hat{\mathcal{G}}_i^2)\|,$$
> >
> > resulting in small $R_{\varepsilon}$. 2) The concentration of augmented representations, where sharper concentration implies larger $\alpha$. Both factors are high-level concepts, thus we heuristically design the new augmentation to meet the requirements, which enhances intra-community edges and purifies inter-community edges.
> >
> > Given some augmentation strategies, we agree that we can design some metrics to measure which augmentation is better based on these bounds. However, if we expect that augmentation design is directly guided by these high-level bounds, finer theoretical analysis is required. Thank you again for your comments and we think it is worth making further study in this direction.

---

> > > ### Comment · Reviewer_5815 · 2022-08-08
> > > **Reply**
> > >
> > > Thank you for the response. I think most of my concerns are addressed. For my first question, it would be good if the authors can provide more evidence on the correlation to make it more convincing. Also, it makes more convincing and clear to explicitly write out the underlying assumption of homophily (or include the results of heterophily and explain why GRADE is not the best choice for heterophilic graph).

---

### Meta-Review · Area_Chair_1itw · 2022-08-27

**Recommendation:** Accept
**Confidence:** Certain

**Metareview:**

This paper identifies a fairness problem in graph contrastive learning (GCL), i.e., GCN often performs bad for low-degree nodes. The key to solve this problem is the observation that GCL can offer more fair representation for both low and high degree nodes. Authors also support their claims with theoretical analysis.

All reviewers appreciate the contributions made by this submission. It is suggested that to simplify notations and make the theorems are self-contained in the final version.

**Award:**

No

---

### Decision · Program_Chairs · 2022-09-14

Accept